# Automatically Inferring Data Quality for Spatiotemporal Forecasting

**Sungyong Seo[1], Arash Mohegh[2], George Ban-Weiss[2], Yan Liu[1]**

[1] Department of Computer Science and [2] Department of Civil and Environmental Engineering
University of Southern California
{sungyons, mohegh, banweiss, yanliu.cs}@usc.edu

## Abstract

Spatiotemporal forecasting has become an increasingly important prediction task in machine learning and statistics due to its vast applications, such as climate modeling, traffic prediction, video caching predictions, and so on. While numerous studies have been conducted, most existing works assume that the data from different sources or across different locations are equally reliable. Due to cost, accessibility, or other factors, it is inevitable that the data quality could vary, which introduces significant biases into the model and leads to unreliable prediction results. The problem could be exacerbated in black-box prediction models, such as deep neural networks. In this paper, we propose a novel solution that can automatically infer data quality levels of different sources through local variations of spatiotemporal signals without explicit labels. Furthermore, we integrate the estimate of data quality level with graph convolutional networks to exploit their efficient structures. We evaluate our proposed method on forecasting temperatures in Los Angeles.

## 1 Introduction

Recent advances in sensor and satellite technology have facilitated the collection of large spatiotemporal datasets. As the amount of spatiotemporal data increases, many have proposed representing this data as time-varying graph signals in various domains, such as sensor networks (Shi et al., 2015; Zhu & Rabbat, 2012), climate analysis (Chen et al., 2014; Mei & Moura, 2015), traffic control systems (Li et al., 2017; Yu et al., 2017), and biology (Mutlu et al., 2012; Yu et al., 2015).

While existing work have exploited both *spatial structures* and *temporal signals*, most of them assume that each signal source in a spatial structure is equally reliable over time. However, a large amount of data comes from heterogeneous sensors or equipment leading to various levels of noise (Song et al., 2015; Zhang & Chaudhuri, 2015). Moreover, the noises of each source can *vary over time* due to movement of the sensors or abrupt malfunctions. This problem raises significantly challenges to train and apply complex *black box* machine learning models, such as deep neural networks, because even a small perturbation in data can deceive the models and lead to unexpected behaviors (Goodfellow et al., 2014; Koh & Liang, 2017). Therefore, it is extremely important to consider data quality explicitly when designing machine learning models.

The definitions of data quality can be varied - high quality data is generally referred to as fitness for intended uses in operations, decision making and planning (Redman, 2008). In this paper, we narrow down the definition as *a penalizing quantity for high local variations*. We consider a learning problem of spatiotemporal signals that are represented by time-varying graph signals for different data qualities. Given a graph $\mathcal{G} = (\mathcal{V}, \mathcal{E}, \mathbf{W})$ and observations $\mathcal{X} \in \mathbb{R}^{N \times M \times T}$ where $N, M, T$ are the number of vertices, the types of signals, and the length of time-varying signals, respectively. We define the concept of data quality levels at each vertex as latent variables, which are connected through a graph using a *local variation* of the vertex. The local variation at each vertex depends on the local spatial structure and neighboring signals. Our definition of data quality can be easily incorporated into any existing machine learning models through a regularizer in their objective functions. In this paper, we develop data quality long short-term memory (DQ-LSTM) neural networks for

spatiotemporal forecasting. DQ-LSTM effectively exploits spatial structures of data quality levels at each vertex through graph convolution, which examines neighboring signals at a set of $K$-hop neighboring vertices, and captures the temporal dependencies of each time series through LSTMs. We demonstrate that data quality is an essential factor for improving the predictive performance of neural networks via experiments on urban heat island prediction in Los Angeles.

**Related work**   A series of work have been conducted on addessing the issues of data qualities and heterogeneous data sources.  Urner et al. (2012) is the first theoretical work that proposes a mixture model for captureing two types of labels in supervised learning. One type of the labels is considered as high quality labels from an expensive source (domain experts) while another type is from error-prone crowdsourcing.  Since the reliability or quality of the labels is different, it is not desired to consider them equally.  The authors proposed a learning algorithm that can utilize the error-prone labels to reduce the cost required for the expert labeling.  Zhang & Chaudhuri (2015) address issues from strong and weak labelers by developing an active learning algorithm minimizing the number of label requests.  Song et al. (2015) focus on the data of variable quality resulting from heterogeneous sources. The authors define the concept of heterogeneity of data and develop a method of adjusting the learning rate based on the heterogeneity. Different from existing works, our proposed framework differentiates heterogeneous sources based on neighborhood signals without any explicit labels.

Another set of work related to our study is learning and processing graph signals or features. Spectral graph theory (Chung, 1997; Von Luxburg, 2007; Shi et al., 2015) has been developed as a main study to understand two aspects of graph signals: structures and signals.  Under this theory many models have been introduced to exploit convolutional neural networks (CNNs) which provide an efficient architecture to extract localized patterns from regular grids, such as images (Krizhevsky et al., 2012).  Bruna et al. (2014) learns convolutional parameters based on the spectrum of the graph Laplacian. Later,  Henaff et al. (2015) extends the spectral aspect of CNNs on graphs into large-scale learning problemsDefferrard et al. (2016) proposes a spectral formulation for fast localized filtering with efficient pooling. Furthermore,  Kipf & Welling (2017) re-formularizes existing ideas into layer-wise neural networks that can be tuned through backpropagation rule with a first-order approximation of spectral filters introduced in Hammond et al. (2011).  Built on these work, we propose a graph convolutional layer that maps spatiotemporal features into a data quality level.

**Outline**   We review graph signal processing to define the local variation and a data quality level (DQL) with graph convolutional networks in Section 2.  In Section 3, we provide how the data quality levels are exploited with recurrent neural networks to differentiate reliability of observations on vertices.  Also, we construct a forecasting model, DQ-LSTM. Our main result is presented in Section 4 with other baselines. In Section 5 we discuss its properties and interpret the data reliability inferred from our model.

## 2   PRELIMINARIES

We first show how to define the local variation at a vertex based on graph signals. Then, we explain how the variational features at each vertex can be used to generate a data quality level.

### 2.1   LOCAL VARIATION

We focus on the graph signals defined on an undirected, weighted graph $\mathcal{G} = (\mathcal{V}, \mathcal{E}, \mathbf{W})$, where $\mathcal{V}$ is a set of vertices with $|\mathcal{V}| = N$ and $\mathcal{E}$ is a set of edges. $\mathbf{W} \in \mathbb{R}^{N \times N}$ is a random-walk normalized weighted adjacency matrix which provides how two vertices are relatively close. When the elements $W_{ij}$ are not be explticly provided by dataset, the graph connectivity can be constructed by various distance metrics, such as Euclidean distance, cosine similarity, and a Gaussian kernel (Belkin & Niyogi, 2002), on the vertex features $\mathbf{V} \in \mathbb{R}^{N \times F}$ where $F$ is the number of the features.

Once all the structural connectivity is provided, the local variation can be defined by the edge derivative of a given graph signal $\mathbf{x} \in \mathbb{R}^N$ defined on every vertex (Zhou & Schölkopf, 2004).

$$\left. \frac{\partial \mathbf{x}}{\partial e} \right|_{e=(i,j)} = \sqrt{W_{ij}}(x(j) - x(i)), \tag{1}$$

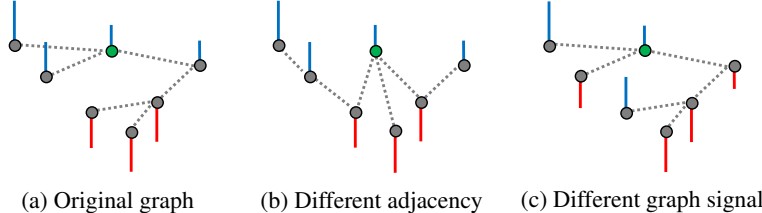

(a) Original graph     (b) Different adjacency     (c) Different graph signal

Figure 1: Each bar represents the signal value at the vertex where the bar originates. Blue and red color indicate positive and negative values, respectively. Local variations at the green node are (a) 1.67 (b) 25 and (c) 15, respectively.

where $e = (i, j)$ is defined as a direction of the derivative and $x(i)$ is a signal value on the vertex $i$. The graph gradient of $\mathbf{x}$ at vertex $i$ can be defined by Eq. 1 over all edges joining the vertex $i$.

$$\nabla_i \mathbf{x} = \left( \left. \frac{\partial \mathbf{x}}{\partial e} \right|_{e=(i,j)} \middle| j \in \mathcal{N}_i \right), \tag{2}$$

where $\mathcal{N}_i$ is a set of neighbor vertices of the vertex $i$. While the dimesion of the graph gradient is different due to the different number of neighbors of each vertex, the local variation at vertex $i$ can be defined by a norm of the graph gradient:

$$\|\nabla_i \mathbf{x}\|_2^2 = \sum_{j \in \mathcal{N}_i} W_{ij}(x(j) - x(i))^2, \tag{3}$$

Eq. 3 provides a measure of local variation of $\mathbf{x}$ at vertex $i$. As it indicates, if all neighboring signals of $i$ are close to the signal at $i$, the local variation at $i$ should be small and it means *less fluctuated signals* around the vertex. As Eq. 3 shows, the local variation is a function of a structural connectivity $\mathbf{W}$ and graph signals $\mathbf{x}$. Figure 1 illustrates how the two factors affect the local variation at the same vertex.

The concept of the local variation is easily generalized to multivariate graph signals of $M$ different measures by repeatedly computing Eq. 3 over all measures.

$$\mathbf{L}_i = (\|\nabla_i \mathbf{x}^1\|_2^2, \cdots, \|\nabla_i \mathbf{x}^m\|_2^2, \cdots, \|\nabla_i \mathbf{x}^M\|_2^2), \tag{4}$$

where $\mathbf{x}^m \in \mathbb{R}^N$ corresponds the $m$th signals from multiple sensors. As Eq. 4 indicates, $\mathbf{L}_i$ is a $M$ dimensional vector describing local variations at the vertex $i$ with respect to the $M$ different measures.

Finally, it is desired to represent Eq. 4 in a matrix form to be combined with graph convolutional networks later.

$$\mathbf{L}(\mathbf{W}, \mathbf{X}) = (\mathbf{D} + \mathbf{W})(\mathbf{X} \odot \mathbf{X}) - 2(\mathbf{X} \odot \mathbf{W}\mathbf{X}), \tag{5}$$

where $\mathbf{D}$ is a degree matrix defined as $D_{ii} = \sum_j W_{ij}$ and $\odot$ is an element-wise product operator. $\mathbf{X}$ is a $N \times M$ matrix describing multivariate graph signals on $N$ vertices and $\mathbf{x}^m$ is a $m$th column of $\mathbf{X}$. $\mathbf{L} \in \mathbb{R}^{N \times M}$ is a local variation matrix and $L_{im}$ is the local variation at the vertex $i$ with respect to the $m$th signal.

## 2.2 DATA QUALITY LEVEL

While the term of data quality has been used in various ways, it generally means "*fitness of use*" for intended purposes (Juran & Godfrey, 1999). In this section, we will define the term under the data property we are interested in and propose how to exploit the data quality level into a general framework.

Given a multivariate graph signal $\mathbf{X} \in \mathbb{R}^{N \times M}$ on vertices represented by a feature matrix $\mathbf{V} \in \mathbb{R}^{N \times F}$, we assume that a signal value at a certain vertex $i$ is desired not be significantly different with signals of neighboring vertices $j \in \mathcal{N}_i$. This is a valid assumption if the signal value at the vertex $i$ is dependent on (or function of) features of the vertex $i$ when an edge weight is defined

by a distance in the feature vector space between two vertices. In other words, if two vertices have similar features, they are connected to form the graph structure and the signal values observed at the vertices are highly likely similar. There are a lot of domains which follow the assumption, for instance, geographical features (vertex features) and meteorological observations (graph signal) or sensory nervous features and received signals.

Under the assumption, we define the data quality level (score) at a vertex $i$ as a function of local variations of $i$:

$$s_i = q(\mathbf{L}_i), \tag{6}$$

It is flexible to choose the function $q$. For example, $s_i$ can be defined as an average of $\mathbf{L}_i$. If so, all measures are equally considered to compute the data quality at vertex $i$. In more general sense, we can introduce parameterized function $q(\mathbf{L}_i; \mathbf{\Phi})$ and learn the parameters through data. Kipf & Welling (2017) propose a method that learns parameters for graph-based features by the layer-wise graph convolutional networks (GCN) with arbitrary activation functions. For a single layer GCN on a graph, the latent representation of each node can be represented as:

$$\mathbf{Z} = \sigma(\hat{\mathbf{A}}\mathbf{X}\mathbf{\Theta}), \tag{7}$$

where $\hat{\mathbf{A}} = (\mathbf{D}+\mathbf{I}_N)^{-\frac{1}{2}}(\mathbf{A}+\mathbf{I}_N)(\mathbf{D}+\mathbf{I}_N)^{-\frac{1}{2}}$ provides *structural connectivities* of a graph and $\mathbf{\Theta}$ is a trainable parameter matrix. $\sigma$ is an activation function. By stacking $\sigma(\hat{\mathbf{A}}\mathbf{X}\mathbf{\Theta})$, it is able to achieve larger receptive fields with multi-layer GCN. See details in Kipf & Welling (2017).

We can replace $\hat{\mathbf{A}}\mathbf{X}$ with $\mathbf{L}$ which is also a function of the weighted adjacency $\mathbf{W}$ and the graph signal $\mathbf{X}$. Note that values in row $i$ of $\hat{\mathbf{A}}\mathbf{X}$ and $\mathbf{L}$ are a function of values at $i$ as well as neighbors of $i$. Although $\mathbf{L}$ only exploits nearest neighbors (i.e., 1-hop neighbors), it is possible to consider $K$-hop neighbors to compute the local variations by stacking GCN before applying Eq. 3. The generalized formula for the data quality level can be represented as:

$$\mathbf{Z} = \sigma_K(\hat{\mathbf{A}}\sigma_{K-1}(\hat{\mathbf{A}}\cdots\sigma_1(\hat{\mathbf{A}}\mathbf{X}\mathbf{\Theta_1})\cdots\mathbf{\Theta}_{K-1})\mathbf{\Theta}_K), \tag{8}$$

$$\mathbf{s} = \sigma_L(\mathbf{L}(\mathbf{W},\mathbf{Z})\mathbf{\Phi}). \tag{9}$$

where $K$ is the number of GCN layers and $\mathbf{s} = (s_1, s_2, \cdots, s_N)$ is the data quality level of each vertex incorporating $K$-hop neighbors. We propose some constraints to ensure that a higher $s_i$ corresponds to *less fluctuated* around $i$. First, we constrain $\mathbf{\Phi}$ to be positive to guarantee that larger elements in $\mathbf{L}$ cause larger $\mathbf{L}\mathbf{\Phi}$ that are inputs of $\sigma_L$. Next, we use an activation function that is inversely proportional to an input, e.g., $\sigma_L(x) = \frac{1}{1+x}$, to meet the relation between the data quality $s_i$ and the local variations $\mathbf{L}_i$. Once $\mathbf{s}$ is obtained, it will be combined with an objective function to assign a penalty for each vertex loss function.

## 3 MODEL

In this section, we give the details of the proposed model, which is able to exploit the data quality defined in Section 2.2 for practical tasks. First, it will be demonstrated how the data quality network DQN is combined with recurrent neural networks (LSTM) to handle time-varying signals. We, then, provide how this model is trained over all graph signals from all vertices.

**Data quality network** In Section 2.2, we find that the local variations around all vertices can be computed once graph signals $\mathbf{X}$ are given. Using the local variation matrix $\mathbf{L}$, the data quality level at each vertex $s_i$ can be represented as a function of $\mathbf{L}_i$ with parameters $\mathbf{\Phi}$ (See Eq. 6). While the function $q$ is not explicitly provided, we can parameterize the function and learn the parameters $\mathbf{\Phi} \in \mathbb{R}^M$ through a given dataset.

One of straightforward parameterizations is based on fully connected neural networks. Given a set of features, neural networks are efficient to find nonlinear relations among the features. Furthermore, the parameters in the neural networks can be easily learned through optimizing a loss function defined for own purpose. Thus, we use a single layer neural networks $\mathbf{\Phi}$ followed by an activation function $\sigma_L(\cdot)$ to transfer the local variations to the data quality level. Note that multi-layer GCN can be used between graph signals and DQN to extract convolved signals as well as increase the

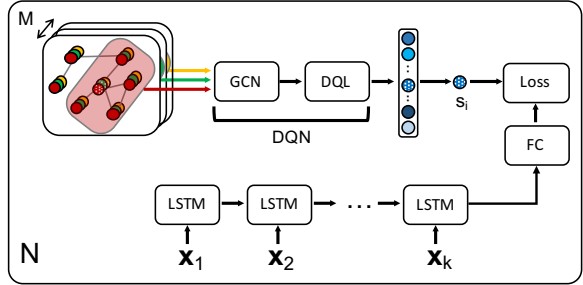

Figure 2: Architecture of DQ-LSTM that consists of GCN followed by the data quality network DQN and LSTM. GCN are able to extract localized features from given graph signals (different colors correspond to different signals) and DQN computes the data quality of each vertex. $N$ is the number of vertices, i.e. each loss function on each vertex is weighted by the quality level $s_i$. Note that the dot-patterned circles denote the current vertex $i$.

size of convolution filters. (See Eq. 8 and 9). This multi-layer neural networks are efficient to learn nonlinear interactions of $K$-hop nodes which are not easily learnable by graph polynomial filters in Sandryhaila & Moura (2013).

**Long short term memory** Recurrent neural networks (RNNs) are especially powerful to extract latent patterns in time series which has inherent dependencies between not only adjacent data points but also distant points. Among existing various RNNs, we use Long short term memory (LSTM) (Hochreiter & Schmidhuber, 1997) to handle the temporal signals on vertices for a regression task. We feed finite lengths $k$ of sequential signals into LSTM as an input series to predict the observation at next time step. The predicted value is going to be compared to a true value and all parameters in LSTM will be updated via backpropagation through time.

**DQ-LSTM** Figure 2 illustrates how the data quality networks with LSTM (DQ-LSTM) consists of submodules. Time-varying graph signals on $N$ vertices can be represented as a tensor form, $\mathcal{X} \in \mathbb{R}^{N \times M \times T}$ where the total length of signals is $T$. First, the time-varying graph signals for each vertex are segmentized to be fed into LSTM. For example, $\mathcal{X}(i, :, h : h + k - 1)$ is one of segmentized signals on vertex $i$ starting at $t = h$. Second, the graph signals for all vertices at the last time stamp $\mathcal{X}(:, :, h + k - 1)$ are used as an input of GCN followed by DQN. Hence, we consider the data quality level by looking at the local variations of the last signals and the estimated quality level $s_i$ is used to assign a weight on the loss function defined on the vertex $i$. We use the mean squared error loss function.

For each vertex $i$, DQ-LSTM repeatedly reads inputs, predicts outputs, and updates parameters as many as a number of segmentized length $k$ time series.

$$\mathcal{L}_i = \frac{1}{n_i} \sum_{j=1}^{n_i} s_i \|\hat{\mathcal{X}}(i, :, k + j - 1) - \mathcal{X}(i, :, k + j - 1)\|_2^2 + \beta \|\mathbf{\Phi}\|_2^2 \tag{10}$$

where $n_i$ is the number of segmentized series on vertex $i$ and $\hat{\mathcal{X}}(i, :, k + j - 1)$ is a predicted value from a fully connected layer (FC) which reduces a dimension of an output vector from LSTM. $L_2$ regularization is used to prevent overfitting. Then, the total loss function over all vertices is as $\mathcal{L} = \frac{1}{N} \sum_{i=1}^{N} \mathcal{L}_i$.

## 4 EXPERIMENTS

In this section, we evaluate DQ-LSTM on real-world climate datasets. In the main set of experiments, we evaluate the mean absolute error (MAE) of the predictions produced by DQ-LSTM over entire weather stations. In addition, we analyze the data quality levels estimated by DQ-LSTM.

### 4.1 Datasets

We use real-world datasets on meteorological measurements from two commercial weather services, Weather Underground(WU) and WeatherBug(WB). Both datasets provide real-time weather information from personal weather stations. In the datasets, all stations are distributed around Los Angeles County, and geographical characteristics of each station are also provided. These characteristics would be used as a set of input features $\mathbf{V}$ to build a graph structure. The list of the static 11 characteristics is, *Latitude, Longitude, Elevation, Tree fraction, Vegetation fraction, Albedo, Distance from coast, Impervious fraction, Canopy width, Building height,* and *Canopy direction.*

Meteorological signals at each station are observed through the installed instruments. The types of the measurements are *Temperature, Pressure, Relative humidity, Solar radiation, Precipitation, Wind speed*, and *Wind direction*. Since each weather station has observed the measurements under its own frequency (e.g., every 5 minutes or every 30 minutes), we fix the temporal granularity at 1 hour and aggregate observations in each hour by averaging them. We want to ensure that the model can be verified and explained physically within one meteorological regime before applying it to the entire year with many other regimes. Since it is more challenging to predict temperatures in the summer season of Los Angeles due to the large fluctuation of daytime temperatures (summer: 36°F / 19°C and winter: 6°F / 3.3°C between inland areas and the coastal Los Angeles Basin), we use 2 months observations from each service, July/2015 and August/2015, for our experiments. The dataset description is provided in Appendix A.

### 4.2 Graph generation

Since structural information between pairs of stations is not directly known, we need to construct a graph of the weather stations. In general graphs, two nodes can be interpreted as similar nodes if they are connected. Thus, as mentioned in Section 2.1, we can compute a distance between two nodes in the feature space. A naive approach to defining the distance is using only the geolocation features, *Latitude* and *Longitude*. However, it might be inappropriate because other features can be significantly different even if two stations are fairly close. For example, the distance between stations in the Elysian Park and Downtown LA is less than 2 miles, however, the territorial characteristics are significantly different. Furthermore, the different characteristics (e.g., Tree fraction or Impervious fraction) can affect weather observations (especially, temperature due to urban heat island effect). Thus, considering only physical distance may improperly approximate the meteorological similarity between two nodes.

To alleviate this issue, we assume that all static features are equally important. This is a reasonable assumption because we do not know which feature is more important since each feature can affect weather measurements. Thus, we normalize all spatial features. In this experiment, we use the Gaussian kernel $e^{(-\gamma\|V_i - V_j\|^2)}$ with $\gamma = 0.2$ and $0.6$ for WU and WB, respectively, and make weights less than 0.9 zero (i.e., disconnected) such that the average number of node neighbors is around 10.

### 4.3 Baselines

We compare our approach to well-studied baselines for time-series forecasting. First, we compare against a stochastic process, autoregressive (AR), which estimates future values based on past values. Second, we further compare against a simple LSTM. This model is expected to infer mixed-dependencies among the input multivariate signals and provide a reference error of the neural networks based model. Lastly, we use graph convolutional networks (Kipf & Welling, 2017) which are also able to infer the data quality level from a given dataset. We test a single layer GCN ($K = 1$) and two-layer GCN ($K = 2$).

### 4.4 Experimental seting

Since DQ-LSTM and our baselines are dependent on previous observations, we set a common lag length of $k = 10$. For the deep recurrent models, the $k$-steps previous observations are sequentially inputted to predict next values. All deep recurrent models have the same 50 hidden units and one fully connected layer ($\mathbb{R}^{50 \times 1}$) that provides the target output. For GCN-LSTM and DQ-LSTM, we

Table 1: Forecasting mean absolute error (MAE) (°C)

|  | AR | LSTM | GCN-LSTM (K=1) | GCN-LSTM (K=2) | DQ-LSTM (K=0) | DQ-LSTM (K=1) |
|---|---|---|---|---|---|---|
| WU7 | 0.5342 | 0.5823 (0.0656) | 0.5152 (0.0081) | 0.5073 (0.0261) | 0.5096 (0.0152) | **0.4788** (0.0111) |
| WU8 | 0.5862 | 0.5911 (0.0221) | 0.5356 (0.0398) | 0.5151 (0.0272) | 0.5087 (0.0117) | **0.4856** (0.0086) |
| WB7 | 0.4812 | 0.4725 (0.0277) | 0.4687 (0.0348) | 0.4411 (0.0321) | 0.4588 (0.0148) | **0.4108** (0.0129) |
| WB8 | 0.5133 | 0.5435 (0.0376) | 0.5412 (0.0483) | 0.5296 (0.0164) | 0.4602 (0.0440) | **0.4574** (0.0178) |

evaluate with different numbers of layers ($K$) of GCN. We set the dimension of the first ($K = 1$) and second ($K = 2$) hidden layer of GCN as 10 and 5, respectively, based on the cross validation. The final layer always provides a set of scalars for every vertex, and we set $\beta = 0.05$ for the $L2$ regularization of the final layer. We use the Adam optimizer (Kingma & Ba, 2015) with a learning rate of 0.001 and a mean squared error objective.

We split each dataset into three subsets: training, validation, and testing sets. The first 60% observations are used for training, the next 20% is used to tune hyperparameters (validation), and the remaining 20% is used to report error results (test). Among the measurements provided, *Temperature* is used as the target measurement, i.e., output of LSTM, and previous time-step observations, including Temperature, are used as input signals. We report average scores over 20 trials of random initializations.

## 5 RESULTS AND DISCUSSION

### 5.1 FORECASTING EXPERIMENT

Experimental results are summarized in Table 1. We report the temperature forecasting mean absolute error (MAE) of our DQ-LSTM model with standard deviations. Meteorological measurements for July and August are denoted by 7 and 8, and $K$ indicates the number of GCN layers. Overall, the models that account for graph structures outperform AR and LSTM. While the node connectivities and weights are dependent on our distance function (Section 4.2), Table 1 clearly shows that knowing the neighboring signals of a given node can help predict next value of the node.

Although GCN are able to transfer a given signal of a vertex to a latent representation that is more compact and expressive, GCN have difficulty learning a mapping from neighboring signals to data quality level directly, unlike DQ-LSTM which pre-transfers the signals to local variations explicitly. In other words, given signal $X$, what GCN learns is $s = f(X)$ where $s$ is the data quality we want to infer from data; however, DQ-LSTM learns $s = g(Y = h(X))$ where $Y$ is a local variation matrix given by $X$ in a closed form, $h$. Thus, lower MAEs of DQ-LSTM verify that our assumption in Section 2.2 is valid and the local variations are a useful metric to measure data quality level. It is also noteworthy that DQ-LSTM with a GCN reports the lowest MAE among all models. This is because the additional trainable parameters in GCN increase the number of neighboring nodes that are accounted for to compute better local variations.

### 5.2 NODE EMBEDDING AND LOW-QUALITY DETECTION

As DQ-LSTM can be combined with GCN, it is possible to represent each node as an embedding obtained from an output of GCN. Embeddings from deep neural networks are especially interesting since they can capture *distances* between nodes. These distances are not explicitly provided but inherently present in data. Once the embeddings are extracted, they can be used for further tasks, such as classification and clustering (Grover & Leskovec, 2016; Kipf & Welling, 2017). Moreover, since the embeddings have low dimensional representations, it is more efficient to visualize the nodes by manifold learning methods, such as t-SNE (Maaten & Hinton, 2008). Visualization with spatiotemporal signals is especially effective to show how similarities between nodes change.

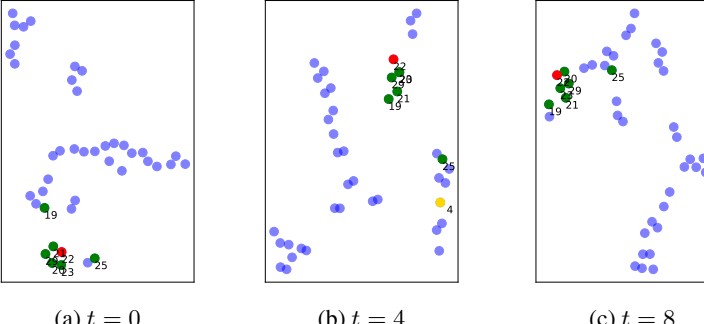

|        (a) $t = 0$        |        (b) $t = 4$        |        (c) $t = 8$        |

Figure 3: t-SNE visualization of outputs of GCN in DQ-LSTM. Red dot denotes the reference node and green dots are the adjacent nodes of the red dot. (a), (b) and (c) illustrate how the embeddings of spatiotemporal signals changes. At $t = 4$, the $v_{25}$ node is relatively far from other green nodes because it is connected with the $v_4$ node which is not a neighbor of red dot. All nodes come from Weather Underground.

Figure 3 shows varying embedding distributions over time. The green dots are neighbors of a red dot, and they are closely distributed to form a cluster. There are two factors that affect embeddings: *temporal signals* and *spatial structure*. Ideally, connected nodes have observed similar signals and thus, they are mapped closely in the embedding space. However, if one node $v_i$ measures a fairly different value compared to other connected values, the node's embedding will also be far from its neighbors. Furthermore, if one node $v_i$ is connected to a subset of a group of nodes $\{g\}$ as well as an additional node $v_j \notin \{g\}$, $v_i$ would be affected by the subset and $v_j \notin \{g\}$ simultaneously. For example, if the signals observed at $v_i$ are similar to the signals at $\{g\}$, the embedding of $v_i$ is still close to those of $\{g\}$. However, if the signals of $v_i$ are close to that of $v_j$, or the weight of $e(i, j)$ is significantly high, $v_i$ will be far away from $\{g\}$ in the embedding space.

Such intuition of the embedding distribution can be used to find potentially low-quality nodes, which we analyze next. Figure 3b shows that a node $v_{25}$ is affected by its neighboring green nodes and $v_4$ that is not included in the cluster (green dots). The red dot $v_{22}$ is connected with the green dots $(v_{19}, v_{20}, v_{21}, v_{23}, v_{25}, v_{29})$. Since these nodes have similar spatial features and are connected, the nodes are expected to have similar observations. At $t = 0$, the distribution of the nodes seems like a cluster. However, $v_{25}$ is far away from other green nodes and the red node at $t = 4$. There are two possible reasons. First, observations of $v_{25}$ at $t = 4$ may be too different with those of other green nodes and the red node. Second, observations at $v_4$, which is only connected to $v_{25}$ (not to other green nodes and the red node), might be too noisy. The first case violates our assumption (See Section 2.2, such that the observation at $v_{25}$ should be similar to those of other green nodes.); therefore, the observations of $v_{25}$ at $t = 4$ might be noisy or not reliable. In the second case, the observations of $v_4$ at $t = 4$ might be noisy. Thus, $v_{25}$ and $v_4$ are candidates of low-quality nodes.

## 5.3 Data quality analysis

Table 2: Observations and inferred DQL

|     | Temp | Press | Humid | DQL |
|-----|------|-------|-------|-----|
| 4   | 28.4 | 1013.1 | 57.7 | 0.022 |
| 19  | 34.5 | 1012.2 | 35.8 | 0.038 |
| 20  | 28.1 | 1011.5 | 58.7 | 0.039 |
| 21  | 28.1 | 1011.5 | 58.7 | 0.039 |
| 22  | 38.1 | 1006.2 | 32.8 | 0.039 |
| 23  | 28.1 | 1011.5 | 58.7 | 0.038 |
| 25  | 28.1 | 1011.5 | 58.7 | 0.030 |
| 29  | 35.3 | 1014.3 | 40.0 | 0.039 |

Since we do not have explicit labels for the data quality levels, it is not straightforward to directly evaluate the data quality inferred from DQ-LSTM. Instead, we can verify the inferred data quality by studying high and low quality examples from embedding distributions and associated meterological observations. Table 2 shows meterological observations associated with the previously discussed embedding distribution at $t = 4$ (Figure 3b). The values $\mathbf{x}_{25}$ at $v_{25}$ are the same as $\mathbf{x}_{20}, \mathbf{x}_{21}$, and $\mathbf{x}_{23}$; however, $\mathbf{x}_{25}$ is quite different than $\mathbf{x}_{19}, \mathbf{x}_{22}$, and $\mathbf{x}_{29}$. Moreover, the edge weights between $v_{25}$ and other green nodes are not as large as weights between other green nodes. ($v_{25}$ is much closer to the ocean than other green nodes.) As a result, it is

not easy for $v_{25}$ to be close to the green nodes ($v_{20}$, $v_{21}$, and $v_{23}$ pull $v_{25}$ and $v_{19}$, $v_{22}$, and $v_{29}$ push $v_{25}$). On the other hand, since $\mathbf{x}_{25}$ is similar to $\mathbf{x}_4$ and $W_{25,4}$ is large, $v_{25}$ is more likely to be close to $v_4$ as in Figure 3b.

Note that $v_4$ has very different geological features compared to the features of the green nodes and thus, $v_4$ is not connected to $v_{22}$ or other green nodes except $v_{25}$. Consequently, $v_{25}$ is the bridge node between $v_4$ and the cluster of $v_{22}$. Since a bridge node is affected by two (or more) different groups of nodes simultaneously, the quality level at the bridge node is more susceptible than those of other nodes. However, this does not directly mean that a bridge node must have a lower data quality level.

As Table 2 shows, $s_4$ has the lowest data quality level, which comes from the discrepancy between its neighboring signals and $\mathbf{x}_4$. Since $v_4$ is connected to $v_{25}$, $v_4$ pulls $v_{25}$ and $s_4$, lowering $s_{25}$ relative to data quality levels of the other green nodes that are correctly inferred.

## 6  CONCLUSION

In this work, we study the problem of data quality for spatiotemporal data analysis. While existing works assume that all signals are equally reliable over time, we argue that it is important to differentiate data quality because the signals come from heterogeneous sources. We proposed a novel formulation that automatically infers data quality levels of different sources and developed a specific formulation, namely DQ-LSTM, based on graph convolution for spatiotemporal forecasting. We demonstrate the effectiveness of DQ-LSTM on inferring data quality and improving prediction performance on a real-world climate dataset. For future work, we are interested in further refining the definitions of data quality and examining rigorous evaluation metrics.

### ACKNOWLEDGMENTS

This work is supported in part by NSF Research Grant IIS-1254206 and IIS-1539608. The views and conclusions are those of the authors and should not be interpreted as representing the official policies of the funding agency, or the U.S. Government. We thank Michael Tsang for discussion and his comments that greatly improved the manuscript.

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

# A  DATASET

In order to have a fair comparison, we use real-world meteorological measurements from two commercial weather service providing real-time weather information, *Weather Underground*(WU)[1] and *WeatherBug*(WB)[2]. Both services use observations from automated personal weather stations (PWS). The PWS are illustrated in Figure 4.

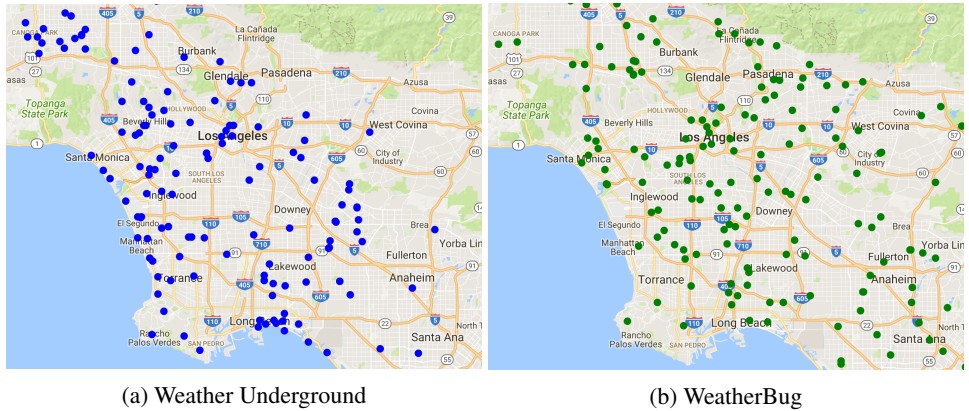

|         (a) Weather Underground          |          (b) WeatherBug          |

Figure 4: Personal weather stations distributed over Los Angeles area

In the dataset, each station is distributed around Los Angeles County and land characteristics where the station is located at are provided. These characteristics would be used as a set of input features, $\mathbf{V}_i$. The list of the static characteristics is:

Table 3: Land features

| Name | Unit | Description |
|------|------|-------------|
| Latitude | degree (°) | An angle which ranges from 0 at the Equator to 90 at the poles |
| Longitude | degree (°) | An angle which ranges from 0 at the Prime Meridian to +180 eastward and 180 westward |
| Elevation | ft | Elevation from average sea level |
| Tree fraction | dimensionless | Fraction covered by tree in that neighborhood |
| Vegetation fraction | dimensionless | Fraction of the neighborhood covered by vegetation |
| Albedo | dimensionless | Reflected amount of incoming shortwave |
| Distance from coast | m | Distance from the nearest coastal point |
| Impervious fraction | dimensionless | Fraction of the neighborhood covered by impervious material |
| Canopy width | ft | Width of the buildings to the centerline of streets |
| Building height | ft | Average height of buildings in neighborhood |
| Canopy direction | degree (°) | The direction of the canopy in degrees from 0-90 |

At each station, a number of weather data are observed through the installed instruments and recorded. The types of measurements are *Temperature, Pressure, Relative humidity, Solar radiation, Precipitation, Wind speed*, and *Wind direction*.

Table 4: Meteorological observations

|  | Temperature | Pressure | Relative Humidity | Solar Radiation | Precipitation | Wind Speed | Wind Direction |
|------|------|------|------|------|------|------|------|
| Unit | °C | mbar | % | W/m$^2$ | mm | km/h | degree |

---

[1] https://www.wunderground.com/

[2] http://weather.weatherbug.com/

