# OpenReview forum: "Automatically Inferring Data Quality for Spatiotemporal Forecasting"
_ICLR.cc/2018/Conference — Accept (Poster)_

### Official Review · AnonReviewer1 · 2017-11-27
**Nice idea and application, but needs significant work before publication.**

**Rating:** 6
**Confidence:** 4

**Review:**

The paper is an application of neural nets to data quality assessment. The authors introduce a new definition of data quality that relies on the notion of local variation defined in (Zhou and Schölkopf, 2004), and they extend it to multiple heterogenous data sources. The data quality function is learned using a GCN as defined in (Kipf and Welling, 2016).

1) How many neighbors are used in the experiments? Is this fixed or is defined purely by the Gaussian kernel weights as mentioned in 4.2? Setting all weights less than 0.9 to zero seems quite abrupt. Could you provide a reason for this? How many neighbors fit in this range?
2) How many data points? What is the temporal resolution of your data (every day/hour/minute/etc.)? What is the value of N, T?
3) The bandwidth of the Gaussian kernel (\gamma) is quite different (0.2 and 0.6) for the two datasets from Weather Underground (WU) and Weather Bug (WB) (sect. 4.2). The kernel is computed on the features (e.g., latitude, longitude, vegetation fraction, etc.). Location, longitude, distance from coast, etc. are the same no matter the data source (WU or WB). Maybe the way they compute other features (e.g., vegetation fraction) vary slightly, but I would guess the \gamma should be (roughly) the same.
4) Are the features (e.g., latitude, longitude, vegetation fraction, etc.) normalized in the kernel? If they are given equal weight (which is in itself questionable) they should be normalized, otherwise some of them will always dominate the distances. If they were indeed normalized, that should be made clear in the paper.
5) Why do you choose the summer months? How does the framework perform on other months and other meteorological signals except temperature? The application is very nice and complex, but I find that the experiments are a little bit too limited.
6) The notations w and W are used for different things, and that is slightly confusing. Usually one is used as a matrix notation of the other.
7) I would tend to associate data quality with how noisy observations are at a certain node, and not heterogeneity. It would be good to add some discussion on noise in the paper. How do you define an anomaly in 5.2? Is it a spatial or temporal anomaly? Not sure I understand the difference between an anomaly and a bridge node.
8) “A bridge node highly likely has a lower data quality level due to the heterogeneity”. I would think a bridge node is very relevant, and I don't necessarily see it as having a lower data quality. This approach seems to give more weight to very similar data, while discarding the transitions, which in a meteorological setting, could be the most relevant ?!
9) Update the references, some papers have updated information (e.g., Bruna et al. - ICLR 2014, Kipf and Welling – ICLR 2017, etc.).

Quality – The experiments need more work and editing as mentioned in the comments.
Clarity – The framework is fairly clearly presented, however the English needs significant improvement.
Originality – The paper is a nice application of machine learning and neural nets to data quality assessment, and the forecasting application is relevant and challenging. However the paper mostly provides a framework that relies on existing work.
Significance – While the paper could be relevant to data quality applications by introducing advanced machine learning techniques, it has limited reach outside the field. Maybe publish it in a data quality conference/journal.

---

> ### Author Response · Authors · 2017-12-26
> **Re: Reviewer 1**
>
> We sincerely thank the reviewer for the insightful comments and suggestions. We would like to stress that assessing and mitigating heterogeneity of data quality is an extremely important but less-studied topic in machine learning. Our work aims at positioning the task and providing one possible solution to address this important problem. The novelty of the paper lies in this important contribution rather than purely the novelty of the proposed model.
>
> Below is our response to the questions.
> 1) and 3) >>
> Thanks for pointing out this potentially confusing issues. To build a symmetric adjacency matrix, we did not set a fixed number of neighbors. Instead, we defined the neighbors by the Gaussian kernel weights with the weight threshold. Since the numbers of the weather stations in WeatherBug(WB) and Weather Underground(WU) are different (#WU:42, #WB:159), the average numbers of neighbors in the datasets are too different under the same bandwidth(\gamma). We aim at evaluating our model under similar degree distribution (similar topology) and therefore, we adjust the bandwidth to make the average numbers of adjacent neighbors in two dataset be similar under the 0.9 threshold. The average degrees are 6.0 and 7.5 for WU and WB, respectively.
>
> 2) >>
> - Each weather station in the data sources (WU and WB) has a different temporal resolution. Some weather stations have observed measurements in every 5 minutes but other weather stations have operated sensors in every 30 minutes. So, we fix the temporal granularity as 1 hour and aggregate observations in each hour by averaging them.
> - N is the total number of weather stations and #WU:42, #WB:159.
> - T is the total length of signals and it is 744(hours) (24(hours/day)*31(day)) for July and August.
>
> 4) >>
> This is a very good point. Yes, we are aware that features represented in a large number (e.g., latitude ~ 34 degree) can dominate features represented in a small number (e.g., vegetation fraction < 1.0) , which is exactly what we want to avoid. Thus, it is a must to normalize the features before using them for computing the distances. As you point out, equal weighting may be not perfectly correct, however, this is what we can do without any domain knowledge.
>
> 5) >>
> - It is a very good point too. Interestingly, the Los Angeles area contains many microclimates, which means that the daytime temperatures can vary as much as 36°F (19°C) between inland areas such as the San Fernando Valley and the coastal Los Angeles Basin. The temperature differences between different areas are more obvious during the summer season (than other seasons). For example, the average high temperature (°F) can vary as much as 26°F for July and August on 9 different regions (Downtown LA, LAX, Santa Monica, Culver City, Long Beach, San Fernando Valley, Burbank, Santa Ana, and Anaheim). In contrast, it varies only 6°F for December and January in the same regions. Thus, it is more challenging to predict temperatures in the summer season, which is why we choose two months (7 and 8) that often shows extreme fluctuations of temperatures.
> - The bigger picture of our work is investigating urban heat island effect in Los Angeles areas. Temperature is the most important factor in urban heat island and exhibits more variations (and more difficult to predict). In contrast,  other observations, such as Relative Humidity or Precipitation, are very stable in Los Angeles areas. They are very easy to predict and therefore cannot justify for a complex model.

---

> > ### Comment · AnonReviewer1 · 2018-01-04
> > **Revised review**
> >
> > I would like to thank the authors for their effort in revising the manuscript. The paper addresses the very important issue of heterogeneity of data sources, however I still believe the manuscript could be further improved to better make the case for this important issue.
> >
> > The explanations added in the paper and in the authors comments are valuable and they should be included as much as possible in future versions of the paper, especially answers to questions 7 and 8 concerning the behavior of bridge nodes and anomalies. The confusing sentence at Q8 has not been changed in the paper, but the authors explanation in the comments is very useful and should be added to the paper. Answer to Q6: I understand the notations—my comment was a suggestion to increase readability of the paper by using different letters for different things, not just lowercase vs uppercase vs bold.
> >
> > I have updated my grading, but still think that the paper needs more work before publication. Forecasting applications are extremely important and challenging, but the experimental section needs to be better explained and further developed. Some of the sentences are still confusing and this makes it hard to follow some of the arguments. Please consider having the manuscript edited by someone who is an English expert.

---

> > > ### Author Response · Authors · 2018-01-05
> > > **Re: Revised review**
> > >
> > > We really appreciate your comments and updated the following.
> > >
> > > "The explanations added in the paper and in the authors comments are valuable and they should be included as much as possible in future versions of the paper, especially answers to questions 7 and 8 concerning the behavior of bridge nodes and anomalies."
> > > >> We added our explanation (Section 5.2 and 5.3) about the behavior of bridge nodes and anomalies in the discussion section (about Q7 and Q8) based on our responses to previous comments. Furthermore, we proofread our discussion to help make the explanation clear.
> > >
> > > "The confusing sentence at Q8 has not been changed in the paper, but the authors explanation in the comments is very useful and should be added to the paper."
> > > >> As you suggested, we updated Section 5.3 to make it clear and less confusing. We added our explanation of how the data quality level is affected by neighboring nodes and quantitatively showed that the level is inferred correctly.
> > >
> > > "Answer to Q6: I understand the notations—my comment was a suggestion to increase readability of the paper by using different letters for different things, not just lowercase vs uppercase vs bold."
> > > >> About Q6: We changed the data quality level notation to $s_i$ instead of $w_i$ for readability. Bold $s$ and $s_i$ are used to point out the data quality level, and $W$ is only used for edge weights.
> > >
> > > "Forecasting applications are extremely important and challenging, but the experimental section needs to be better explained and further developed. Some of the sentences are still confusing and this makes it hard to follow some of the arguments. Please consider having the manuscript edited by someone who is an English expert."
> > > >> We added details of our experiments (including experimental setting, discussion of results) and edited the manuscript carefully (especially the Abstract, Introduction, Experiments, Results, and Conclusion sections) with the help of a native English speaker to improve its readability.
> > >
> > > Thank you for your comments

---

> ### Author Response · Authors · 2017-12-26
> **Re: Reviewer 1**
>
> 6) >>
> We use the lowercase w for data quality levels for each station and the upper case W for edge weights. The bold w is the vector having N elements and w_i is the data quality level of an i-th weather station. We have fixed the confusions in the updated draft.
>
> 7) >>
> - In our paper, we assume that if two connected nodes (weather stations) are located in similar spatial features, these nodes highly likely observe similar meteorological measurements. Thus, if two connected nodes provide very different observations, we think that these observations are not reliable and could be noisy. Furthermore, if there is a group of connected nodes and some nodes observe significantly different measurements with other nodes in the group, we can say that the measurements are too heterogeneous and not reliable. In other words, the data quality level is associated with noisy as well as heterogeneous observations.
> - For the anomaly detection, what we would like to propose is that the embeddings from our model can be used to visualize nodes based on their connectivity and observations. For example, in Figure 3, the red dot (v_22) is connected with the green dots (v_19, v_20, v_21, v_23, v_25, v_29). Since these nodes have similar spatial features and are connected, it is expected to have similar observations. At t=0, the distribution of the nodes seems like a cluster. However, v_25 is far away from other green nodes and the red node at t=4. There are two possible reasons. First, observations of v_25 at t=4 may be too different with those of other green nodes and the red node. Second, observations of a node (v_4) that is only connected to v_25 (not other green nodes and the red node) might be too noisy. In the first case, since it violates our assumption (v_25's observations should be similar with those of another green node.), the observations of v_25 at t=4 might be noisy or not reliable. In the second case, the observations of v_4 at t=4 might be noisy. Thus, we can detect potentially anomalous nodes by looking at the distribution of nodes. (A bridge node is not necessary to be an anomaly.)
> - The anomaly comes from temporal observations by comparing to neighbor observations. Thus, two aspects (spatial and temporal) are jointly considered.
>
> 8) >>
> Thanks for your pointing out. We agree that the sentence you quote can cause some confusion. A bridge node in our paper is considered as a node connecting two (or more) clusters that consist of nodes having similar features. As a result, a bridge node is affected by two (or more) different groups of nodes simultaneously, and thus, the quality level at the bridge node is more susceptible than those of other nodes. However, it doesn’t directly mean that it is necessary for a bridge node to have a lower data quality level.
>
> 9) >>
> Thanks for your suggestion and we update the conference information in the revised version.
>
> Quality, Clarity, Originality, Significance >>
> We do not agree with the comment on significance. Our paper proposes a novel solution to automatically infer data quality levels based on local variations of graph signals and demonstrate that the quality levels can be used to reduce forecasting error and detect potentially anomalous observations. It provides a new idea on inferring interpretable quantity without explicit labels. We agree that there is limited work in ICLR  on data quality, but it is definitely one essential hurdle for any representation learning model to work in practice.

---

### Official Review · AnonReviewer3 · 2017-11-27
**Interesting work. Good novelty component. Uncertain about practical impact.**

**Rating:** 6
**Confidence:** 3

**Review:**

Update:

I have read the rebuttal and the revised manuscript. Paper reads better and comparison to Auto-regression was added. This work presents a novel way of utilizing GCN and I believe it would be interesting to the community. In this regard, I have updated my rating.

On the downside, I still remain uncertain about the practical impact of this work. Results in Table 1 show that proposed method is capable of forecasting next hour temperature with about 0.45C mean absolute error. As no reference to any state of the art temperature forecasting method is given (i.e. what is the MAE of a weather app on a modern smartphone), I can not judge whether 0.45C is good or bad. Additionally, it would be interesting to see how well proposed method can deal with next day temperature forecasting.

---------------------------------------------
In this paper authors develop a notion of data quality as the function of local variation of the graph nodes. The concept of local variation only utilizes the signals of the neighboring vertices and GCN is used to take into account broader neighborhoods of the nodes. Data quality then used to weight the loss terms for training of the LSTM network to forecast temperatures at weather stations.

I liked the idea of using local variations of the graph signals as quality of the signal. It was new to me, but I am not very familiar with some of the related literature. I have one methodological and few experimental questions.

Methodology:
Why did you decide to use GCN to capture the higher order neighborhoods? GCN does so intuitively, but it is not clear what exactly is happening due to non-linearities. What if you use graph polynomial filter instead [1] (i.e. linear combination of powers of the adjacency)? It can more evidently capture the K-hop neighborhood of a vertex.

Experiments:
- Could you please formalize the forecasting problem more rigorously. It is not easy to follow what information is used for training and testing. I'm not quite certain what "Temperature is used as a target measurement, i.e., output of LSTM, and others including Temperature are used as input signals." means. I would expect that forecasting of temperature tomorrow is solely performed based on today's and past information about temperature and other measurements.
- What are the measurement units in Table 1?
- I would like to see comparison to some classical time series forecasting techniques, e.g. Gaussian Process regression and Auto-regressive models. Also some references and comparisons are needed to state-of-the-art weather forecasting techniques. These comparisons are crucial to see if the method is indeed practical.

Please consider proofreading the draft. There are occasional typos and excessively long wordings.

[1] Aliaksei Sandryhaila and José MF Moura. Discrete signal processing on graphs. IEEE transactions
on signal processing, 61(7):1644–1656, 2013.

---

> ### Author Response · Authors · 2017-12-26
> **Re: Reviewer 3**
>
> Thank you for your comments and suggestions to improve the paper. Below are our responses to the main points of your comments:
>
> Methodology
> >> It is commonly believed that the earth climate system is a complex one involving many nonlinear interactions between climate factors. Neural networks have proven to be an effective solution to capture nonlinear dependencies in many applications. To capture nonlinearity, we use a nonlinear activation function (ReLU). Furthermore, multiple layers are more effective to learn features at various levels of abstraction.
> Graph polynomial filter (Equation (7) in [1]) has the similar form as GCN filter operations, but it does not consider nonlinearity. GCN is also based on the polynomial of the adjacent matrix, it equivalently handles the K-hop neighborhood of a vertex.
>
> Experiments
> >> Could you please formalize the forecasting problem more rigorously.
> Thanks for pointing out the confusion. For the sentence you quoted, we did intend to describe as your suggested - In other words, future temperatures are forecasted by looking at past temperatures as well as other meteorological observations. We have updated the expression more clearly.
>
> >> What are the measurement units in Table 1?
> We have added the details of the climate dataset in the appendix in the updated draft.
>
> >>  I would like to see comparison to some classical time series forecasting techniques, e.g. Gaussian Process regression and Auto-regressive models.
> Thanks for the baseline suggestions. We have added the results by auto-regressive for robust comparison in the updated draft.
>
> Thanks for your comments, and we proofread the draft and make it more clear.

---

> > ### Comment · AnonReviewer3 · 2018-01-14
> > **Measurement units in Table 1**
> >
> > Thank you for clarifications and improving the draft. Although I am still not certain about the units in Table 1. In the Appendix you mentioned that temperature is in C, however you did not update Table 1. Is MAE given in Celsius? For the training purposes features and response are often normalized or pre-processed somehow, therefore I want to confirm that the results you report correspond to actual Celsius.

---

> > > ### Author Response · Authors · 2018-01-14
> > > **Re: Measurement units in Table 1**
> > >
> > > Thanks for your pointing out.
> > >
> > > Yes, the unit in Table 1 is Celcius (C). As you mentioned, we normalize features and responses, and feed them into our model. Once we get the predicted values (which are in a normalized range) from our model, we do denormalization (inverse normalization) to recover the Celcius unit to report MAE results.
> > > The predicted values are denormalized by the constants (standard deviation and mean) from the training dataset.

---

> > > > ### Comment · AnonReviewer3 · 2018-01-14
> > > > **Thanks**
> > > >
> > > > Thank you for clarifying. Please consider adding units to the title of Table 1.
> > > >
> > > > I have updated my review and rating.

---

### Official Review · AnonReviewer2 · 2017-11-28
**This work proposed a new way to evaluate the quality of different data sources with the time-vary graph model, that is, the quality of data sources (e.g., time-series sequences) associated with each vertex is defined as a function of local variation at each vertex. In addition, the quality level is also used as a regularization term in the objective function for any general applications. The data quality evaluation is trained using the Multi-layer graph convolutional networks.**

**Rating:** 8
**Confidence:** 4

**Review:**

Summary of the reviews:
Pros:
•	A novel way to evaluate the quality of heterogeneous data sources.
•	An interesting way to put the data quality measurement as a regularization term in the objective function.
Cons:
•	Since the data quality is a function of local variation, it is unclear about the advantage of the proposed data quality regularization versus using a simple moving average regularization or local smoothness regularization.
•	It is unclear about the advantage of using the Multi-layer graph convolutional networks versus two naïve settings for graph construction + simple LSTM, see detailed comments D1

Detailed comments:
D1: Compared to the proposed approaches, there are two alternative naïve ways: 1) Instead of construct the similarity graph with Gaussian kernel and associated each vertex with different types of time-series, we can also construct one unified similarity graph that is a weighted combination of different types of data sources and then apply traditional LSTM; 2) During the GCN phases, one can apply type-aware random walk as an extension to the deep walk that can only handle a single source of data. It is unclear about the advantage of using the Multi-layer graph convolutional networks versus these two naïve settings. Either some discussions or empirical comparisons would be a plus.

---

> ### Author Response · Authors · 2017-12-26
> **Re: Reviewer 2**
>
> Thank you for your comments and suggestions to improve the paper. Below are our responses to the main points of your comments:
>
> Cons 1.
> >> It is an interesting point.  A simple moving average or local smoothness regularization may improve the forecasting performance. However, these regularizations cannot infer the data quality level which is useful to understand time-varying graph signals.
>
> Cons 2 and Detailed comments
> >> Thanks for the suggestions of more baselines. For the former one, we do not have a specific way to get the unified similarity graph based on the different types of time-series. That is why we only consider the spatial (static) features to construct the graph structure. But it could be an interesting direction to explore.
> It is a great idea to think different ways to cover K-hop neighbor nodes. Random deep walk is one of the ways. Although we haven’t compared our method to these random-walk-based methods, the multi-layer GCNs with the data quality networks are more flexible to learn latent interactions between temporal observations due to their nonlinearity and compositional property. It may be possible to improve forecasting quality by using the random walk method, however, we focus on the automatically inferring method of the data quality level, which is not easily achievable by other ways.

---

### Decision · Program_Chairs · 2018-01-29
**ICLR 2018 Conference Acceptance Decision**

**Decision:**

Accept (Poster)

**Comment:**

With an 8-6-6 rating all reviewers agreed that this paper is past the threshold for acceptance.

The  quality of the paper appears to have increased during the review cycle due to interactions with the reviewers. The paper addresses issues related to the quality of heterogeneous data sources. The paper does this through the framework of graph convolutional networks (GCNs). The work proposes a data quality level concept defined at each vertex in a graph based on a local variation of the vertex. The quality level is used as a regularizer constant in the objective function. Experimental work shows that this formulation is important in the context of time-series prediction.

Experiments are performed on a dataset that is less prominent in the ML and ICLR community, from two commercial weather services Weather Underground and WeatherBug; however, experiments with reasonable baseline models using a "Forecasting mean absolute error (MAE)" metric seem to be well done.

The biggest weakness of this work was a lack of comparison with some more traditional time-series modelling approaches. However, the authors added an auto-regressive model into the baselines used for comparison. Some more details on this model would help.

I tend to agree with the author's assertion that: "there is limited work in ICLR  on data quality, but it is definitely one essential hurdle for any representation learning model to work in practice. ".

For these reasons I recommend a poster.